METHODS AND RESOURCES

# Day-night and seasonal variation of human gene expression across tissues

**Valentin Wucher**[1,2,3,4‡], **Reza Sodaei**[1‡], **Raziel Amador**[1], **Manuel Irimia** [1,5,6]*, **Roderic Guigó** [1,5]*

1 Centre for Genomic Regulation, Barcelona Institute of Science and Technology, Barcelona, Spain,
2 MeLiS, SynatAc Team, UCBL1—CNRS UMR5284—Inserm U1314, Lyon, France, 3 French Reference Center on Paraneoplastic Neurological Syndrome, Hospices Civils de Lyon, Lyon, France, 4 University of Lyon, Université Claude Bernard Lyon 1, Lyon, France, 5 Universitat Pompeu Fabra, Barcelona, Spain, 6 ICREA, Barcelona, Spain

‡ These authors share first authorship on this work.
* mirimia@gmail.com (MI); roderic.guigo@crg.eu (RG)

## Abstract

Circadian and circannual cycles trigger physiological changes whose reflection on human transcriptomes remains largely uncharted. We used the time and season of death of 932 individuals from GTEx to jointly investigate transcriptomic changes associated with those cycles across multiple tissues. Overall, most variation across tissues during day-night and among seasons was unique to each cycle. Although all tissues remodeled their transcriptomes, brain and gonadal tissues exhibited the highest seasonality, whereas those in the thoracic cavity showed stronger day-night regulation. Core clock genes displayed marked day-night differences across multiple tissues, which were largely conserved in baboon and mouse, but adapted to their nocturnal or diurnal habits. Seasonal variation of expression affected multiple pathways, and it was enriched among genes associated with the immune response, consistent with the seasonality of viral infections. Furthermore, they unveiled cytoarchitectural changes in brain regions. Altogether, our results provide the first combined atlas of how transcriptomes from human tissues adapt to major cycling environmental conditions. This atlas may have multiple applications; for example, drug targets with day-night or seasonal variation in gene expression may benefit from temporally adjusted doses.

## Introduction

The yearly and daily movement of the earth around the sun and around itself has created a continuously changing environment since the origin of life to which all organisms across the phylogenetic spectrum have adapted. In mammals, in particular, behavioral adaptations to daily rhythm include the regulation of sleep and feeding cycles. Recent studies have investigated how the physiological responses to the daily cycle are reflected at the transcriptomic level. These studies have reported large-scale circadian gene expression oscillations in mice [1], baboons [2], and humans [3,4]. Unlike the circadian process, circannual rhythm in mammals has been less studied. Animals exhibit a range of behavioral and physiological adaptations in different seasons, such as hibernation and alterations of the coat color in polar animals

**Data Availability Statement:** All GTEx open-access data are available on the GTEx Portal (https://gtexportal.org/home/datasets). All GTEx protected data are available via dbGaP (accession phs000424.v8). Drugs and their target genes were

downloaded from the DGI database (https://dgidb. org/downloads, version 2022-Feb). Supp Datasets 1 and 2 can be accessed through figshare at https://doi.org/10.6084/m9.figshare.21906252.v1 and https://doi.org/10.6084/m9.figshare. 21906255.v1, respectively. All other relevant data are within the paper and its Supporting Information files. All the data underlying each figure can be found in the S1 Data file.

**Funding:** The research reported in this publication was supported by the National Human Genome Research Institute of the National Institutes of Health (R01MH101814 and 5U24HG009446 to R. G.), by the Spanish Ministry of Science and Innovation (PGC2018-094017-B-I00 to R.G. and BFU2017-89201-P to M.I.), and by the European Research Council (ERC) under the European Union's Horizon 2020 research and innovation program (ERC-StG-LS2-637591 and ERC-CoG-LS2-101002275 to M.I.). A.R. was a predoctoral fellow of the CONACYT "Becas al Extranjero" Program of Mexico. We acknowledge the support of the Spanish Ministry of Science and Innovation to the EMBL partnership, Centro de Excelencia Severo Ochoa and CERCA Programme / Generalitat de Catalunya. The funders had no role in study design, data collection and analysis, decision to publish, or preparation of the manuscript.

**Competing interests:** The authors have declared that no competing interests exist.

**Abbreviations:** BMI, body mass index; COVID-19, Coronavirus Disease 2019; DGIdb, Drug–Gene Interaction database; FDR, false discovery rate; GO, Gene Ontology; SARS-CoV-2, Severe Acute Respiratory Syndrome Coronavirus 2; TPM, transcripts per million.

[5,6]. One way in which mammals regulate their seasonal reproductive behavior, growth, food intake, and migratory behavior is via the brain-gonadal and other hormonal axes [7–9]. In humans, numerous pathologies present a strong seasonal pattern, which is particularly prominent for many infectious diseases, but also observed in complex cardiovascular and psychiatric disorders [10–13]. Nevertheless, despite its relevance for human physiology and disease, genome-wide studies on circannual rhythms are scarce. Castro Dopico and colleagues (2015) analyzed the transcriptome of white blood cells from children in Germany, from individuals affected with type 1 diabetes from the United Kingdom, and from asthmatic cohorts from distinct geographical locations [14]. They found many genes with seasonal expression profiles, inverted between Europe and Oceania. These seasonal expression profiles were prominent in genes from the immune system. To our knowledge, however, there are no genome-wide studies of the transcriptional impact of these adaptations to seasonal variation across multiple human tissues.

Here, we leverage on the deep transcriptome data across human tissues produced by the GTEx consortium (16,151 RNA-seq samples of 932 postmortem human donors from 46 tissues) [15] to investigate the transcriptional impact of circadian and circannual rhythms in an unprecedented number of tissues. GTEx transcriptional measurements are taken exclusively at the donor's death; therefore, there is a single time point measure per individual. In addition, GTEx metadata only includes the time of the day and the season of death, but not the actual day, the week, or even the month of death. This prompted us to artificially discretize circadian and circannual variation into day-night and season-specific variation. Despite all these caveats, we show that, when aggregated over many individuals, these transcriptional snapshots randomly distributed along time create temporal trajectories that recapitulate day-night and seasonal transcriptional variation, and they constitute, therefore, a unique resource to investigate this variation.

## Results

### Tissue-dependent day-night variation in gene expression

We first used MetaCycle [16], a suite designed to analyze rhythmic data and that has been previously used to investigate circadian patterns (e.g., Ruberto and colleagues [17] and Mishra and colleagues [18]). While MetaCycle uses three methods to identify oscillating genes, only Lomb-Scargle is suited for GTEx data, due to the uneven distribution of the time of death. Using this method, from the 18,018 protein-coding genes expressed in at least one tissue (median transcripts per million (TPM) $\geq 1$), we identified 187 (1%) that were circadian in at least one tissue (non-adjusted $P \leq 0.05$; S1 Table). This number is much smaller than that previously reported in baboon [2], where 82% of the protein-coding genes had circadian patterns in at least one of the 64 studied tissues. Lack of power to detect genes with circadian gene expression patterns can be partially attributed to the characteristics of the GTEx resource. As reported, the GTEx metadata only includes the time of the day and the season of death, and there is no information about the location of death. Therefore, we focused only on the individuals in which the time of death could be unequivocally classified as either day [8:00 to 17:00] (351 individuals) or night [21:00 to 5:00] (315 individuals) and excluded those in which death occurred outside these intervals (uncertain data points, referred to hereafter as the twilight; 222 individuals) (Fig 1A). We then identified genes that were differentially expressed between day and night.

For this purpose, we carried out a differential gene expression analysis using *voom/limma* [19] controlling for the effects of season of death, sex, body mass index (BMI), age, and postmortem interval. We performed an initial comparison across tissues with high sensitivity,

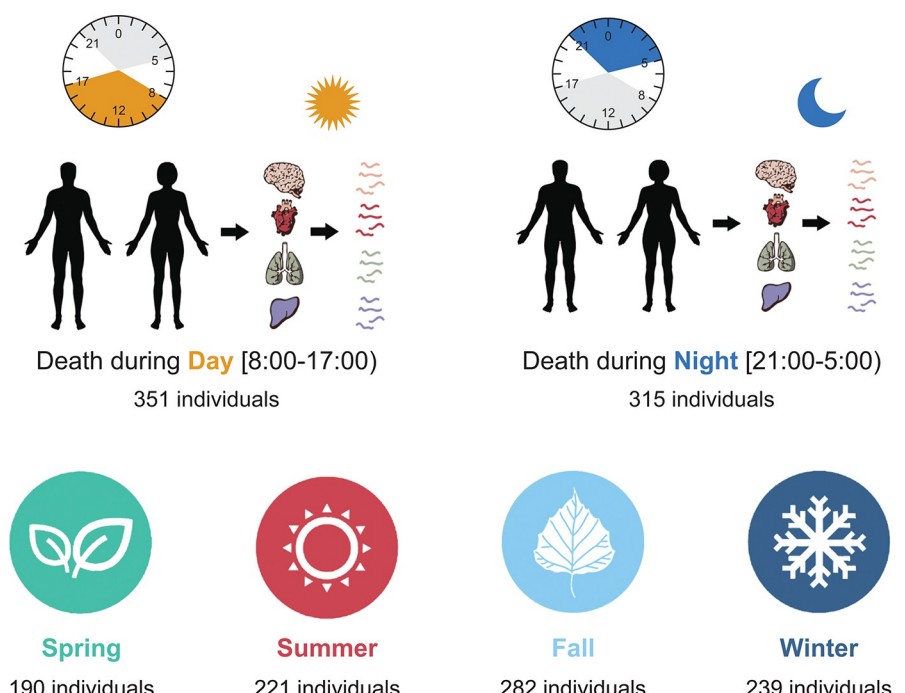

**Fig 1. Day-night and seasonal classification of the GTEx samples.** (**A**) Samples from GTEx were classified as either day or night depending on the time of death of the donor, respectively, between 08:00 and 17:00 (351 individuals) and between 21:00 and 05:00 (315 individuals). (**B**) Samples from GTEx were assigned to the different seasons (spring, summer, fall, and winter), according to the reported season of death of the donor.

applying loose cutoffs of a non-adjusted $P \leq 0.05$ and an absolute $\log_2$ fold-change $\geq 0.1$. From the 18,022 protein-coding genes expressed in at least one tissue (median TPM $\geq 1$), we found that 12,530 (70%) were differentially expressed between day and night in at least one of the 46 tested tissues (day-night genes, Supp Dataset 1 - https://doi.org/10.6084/m9.figshare.21906252.v1), which is a number in line with the results found in baboon [2], with a Simpson index of similarity of 0.7 between our day-night genes and the baboon circadian genes (see Methods and S2 Table). Using a more stringent cutoff, with false discovery rate (FDR) $\leq 0.1$, we identified 4,928 (27%) of genes as differentially expressed between day and night (Supp Dataset 1 - https://doi.org/10.6084/m9.figshare.21906252.v1).

Per tissue, 5.5% of expressed protein-coding genes were day-night on average, using the most sensitive settings (non-adjusted $P \leq 0.05$ and an absolute $\log_2$ fold-change $\geq 0.1$). The tissue with the largest number of day-night genes was lung, with 2,418 genes (17.2% of genes expressed in this tissue; Fig 2A). Other tissues with a proportionally large number of day-night genes were the heart left ventricle (2,202 genes, 19.2%) and whole blood (1,900 genes, 19%). On the other hand, the tissue with the fewest differentially expressed day-night genes was salivary gland, with only 85 genes (0.63% of genes expressed in this tissue; Fig 2A). Other tissues with proportionally low number of day-night genes were colon transverse (92, 0.67%) and testis (105, 0.66%). Brain regions also showed a relatively small number of day-night changes (ranging from 0.86% to 7.8%; Fig 2A). Caudate was the brain region with the highest number of day-night genes (1,026 7.8%), followed by the cerebellum (766, 5.7%), which was reported to have a sleep stage–dependent activity [20]. Using FDR $\leq 0.1$ as a cutoff for the differential expression analysis showed fewer differentially expressed genes for all tissues, but the tissues with the highest relative numbers were maintained (S1 Fig). Moreover, randomizations by

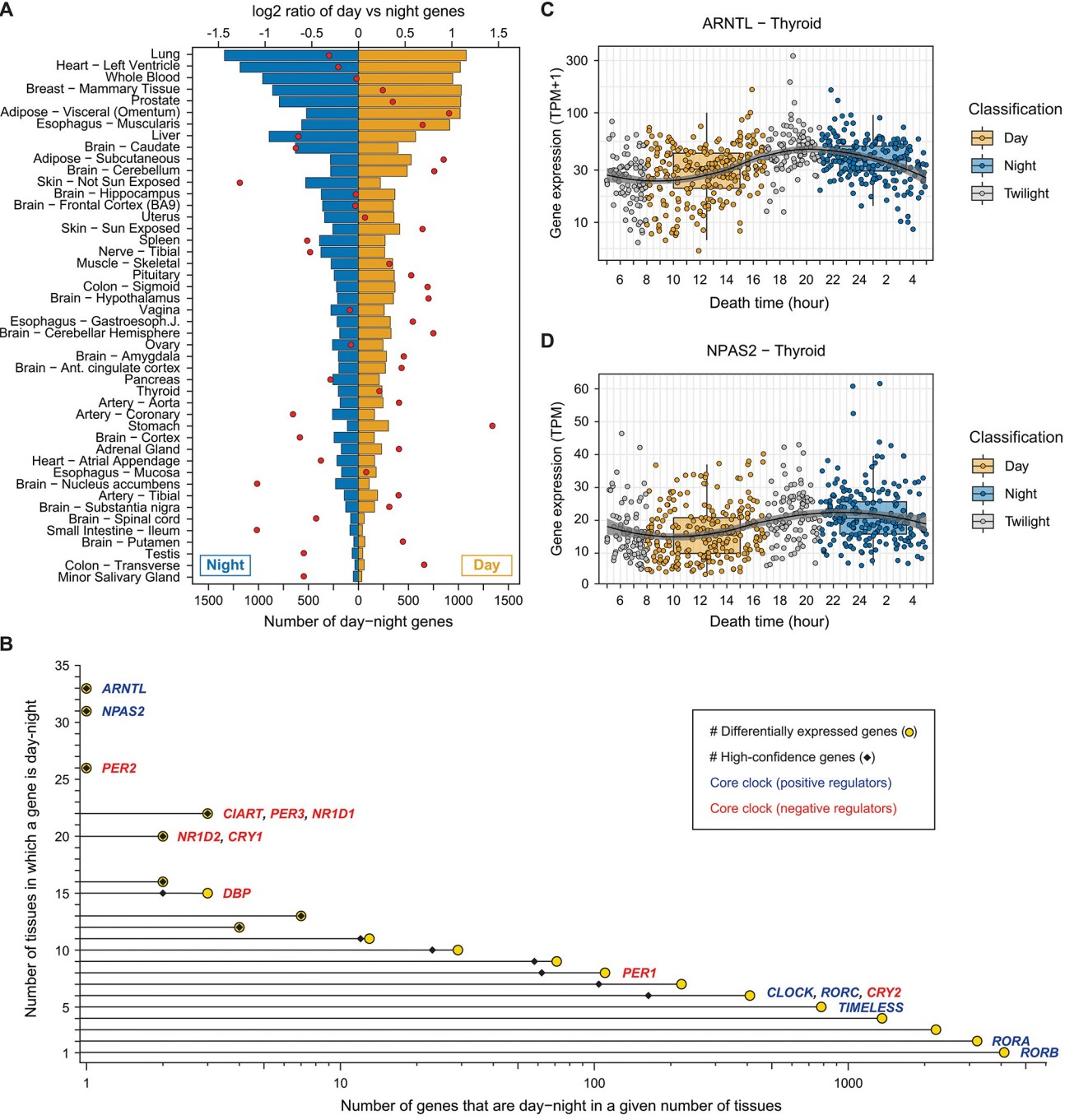

**Fig 2. Distribution of the day-night genes in the GTEx tissues.** (**A**) Number of genes found as day-night, i.e., genes differentially expressed between day and night (barplots; bottom x-axis), and log$_2$ ratio between the number of genes up-regulated during the day vs. the night (red dots; top x-axis) for each tissue. Tissues are sorted by the total number of day-night genes. (**B**) Yellow dots represent the number of tissues in which a given gene is classified as day-night (y-axis) vs. the number of genes that are classified as day-night in that number of tissues (x-axis, log$_{10}$ scale). Gene names for core clock genes are shown next to the dot with the corresponding number of tissues in which they are identified as day-night. Black diamonds show the number of high-confidence day-night genes per number of tissues (see S5 Fig for further details). (**C**) Expression of *ARNTL* in the thyroid in GTEx samples at the time of death of the GTEx donors (in hours). The colors of the dots represent the classification of the individuals according to their time of death: during the day (yellow), during the night (blue), or during twilight (grey). The samples classified as twilight were discarded for the day-night analysis. The "circadian" curve was created using the *geom_smooth* function from *ggplot2* in R with the "loess" method. The data underlying this figure can be found in S1 Data.

swapping the times of death of individuals within each tissue (see Methods) showed that these relative patterns were not due to differences in sample size and/or the structure of samples within each tissue (S2 Fig), and the top tissues displayed a strong enrichment in the number of differentially expressed genes between day and night compared to the randomized samples. Finally, some tissues showed a bias in the number of genes overexpressed during the day (diurnal genes) versus the night (nocturnal genes) (Fig 2A and S3 Table). Stomach was the tissue with the strongest diurnal preference, while non-sun-exposed skin was the tissue with the strongest nocturnal preference. In contrast, sun-exposed skin showed diurnal preference (0.7 $\log_2$ day/night genes ratio), suggesting that UV plays a role in the activation of gene expression.

Importantly, while our approach increases the power to detect significant changes in gene expression between day and night, it does not necessarily lead to the identification of all genes with circadian patterns of gene expression, since the circadian peaks may occur at the twilight time zones that we are ignoring. Consistently, most of the gene-tissue pairs detected by Meta-Cycle as circadian were also identified as day-night by our analyses (210/339, 62%; S1 Table), with most exceptions peaking at or around the twilight (S4 Table). To further understand the contribution of day-night variation to global circadian variation in gene expression, we implemented a sliding window approach in which we identified all genes that were differentially expressed between pairs of 6-hour window intervals separated by 6 hours (see Methods). Aggregating the results across all tissues, and despite the differences in the time interval definitions, the largest number of differentially expressed genes between two specific time windows was obtained for 7/8 AM to 1/2 PM versus 7/8 PM to 1/2 AM, highly overlapping with our day/night (S3A Fig). This suggests that day-night variation is, as expected, a major contributor to circadian-like variation in gene expression. Moreover, there was a strong and significant correspondence between the number of genes that were differentially expressed between at least one pair of time windows (to which we collectively refer as "circadian-like") and day-night genes across tissues (S3B Fig; Spearman rho = 0.56, $P = 0$).

## Recurrent day-night variation in gene expression across tissues

We next computed the number of tissues in which a given gene showed a day-night pattern (Fig 2B). We found that, on average, day-night genes exhibited a day-night pattern in 2.6 tissues out of 39 tissues in which they had detectable expression (median TPM $\geq$ 1), indicating a rather tissue-specific response to day-night cycles. Overall, we found that genes identified as day-night in more than one tissue significantly tended to be consistently up-regulated in either day or night, and this consistency largely increased with the number of tissues in which genes were detected as day-night (S5 Table; see Methods).

We focused on a set of 16 genes that form the molecular core clock and are the main regulators of the circadian rhythm [21–23] (S6 Table). With the exception of *RORC* and *RORB*, the core clock genes were expressed in almost all tissues, and, with the exception of *RORB*, they showed differential day-night expression in multiple tissues (Fig 2B). This is in agreement with the hypothesis that one molecular clock program is present in all tissues but the circadian processes triggered downstream are highly tissue-specific [24]. Core clock genes usually have effect sizes (difference between day and night expressions) that are larger than those of non-core clock day-night genes (S4 Fig). *ARNTL*, a positive regulator of the core clock, was the gene with a day-night pattern in the largest number of tissues (33 tissues; Fig 2B). Its expression at the time of death, aggregated across the GTEx donors, nicely captures the known circadian behavior of this gene (Fig 2C), strongly indicating that, in spite of all the caveats associated with the data collection and available metadata, the GTEx data can be effectively

used to investigate gene expression patterns during the day-night cycle. A recent study described *ARNTL* as the main regulator of the intertissue timekeeping function in mouse [25], and our results suggest that it may play a similar role in humans. Similarly, *NPAS2* was the second gene with a day-night pattern in the largest number of tissues (31 tissues; Fig 2B and 2D) and has also been reported as being circadian in many mouse [26] and baboon [2] tissues.

Among the core clock genes, the thyroid had the highest number of day-night cases (12 out of 16 genes; Fig 3A). In contrast, we did not detect any of them as day-night in stomach, testis, and vagina, consistent with their overall low number of day-night changes (Fig 2A) and with previous studies in testis in rodents [27]. Clock genes showed a largely consistent pattern of day-night expression across tissues (i.e., they were either consistently up- or down-regulated at the same time of the day; Fig 3A). One exception was *NR1D2*, whose expression was consistently higher diurnally in brain subregions but nocturnally in all other tissues (Figs 3B and S5). For the core clock genes that showed day-night differences in both human and baboon in any of the available 20 homologous tissues (S2 Table), we found that most orthologs had a similar behavior in both species (59 similar versus 15 opposite gene-tissue pairs; $P = 1.277 \times 10^{-7}$ one-sided binomial test; Fig 3C), consistent with their shared diurnal regimes. In contrast, core clock genes in mouse (a preferentially nocturnal mammal) largely showed the opposite behavior than their human counterparts (13 similar versus 38 opposite gene-tissue pairs; $P = 3.105 \times 10^{-4}$ one-sided binomial test; Fig 3C and S7 Table).

Next, we aimed at identifying a set of "high-confidence day-night genes" from which to obtain further functional insights. Although the highly sensitive approach that we implemented above is well suited to reveal tissue-level patterns, it might be oversensitive to generate reliable individual gene-level insights. Therefore, we defined as high-confidence day-night genes a set of 445 genes with a consistent day (282) or night (165) pattern in multiple tissues ($P \leq 0.05$ two-sided binomial test across all tissues, brain regions, and/or across non-brain tissues; see Figs 2B and S6 and S8 Table and Methods). High-confidence day-night genes included most clock genes (with the exception of *TIMELESS*, *RORA*, and *RORB*) and were significantly enriched for circadian and photoperiodism-related Gene Ontology (GO) terms (Fig 3D and S9 Table). Moreover, this gene set significantly overlapped those annotated as circadian in the Circadian Gene DataBase in humans [28] (87 genes, $P = 3.074 \times 10^{-8}$ one-sided Fisher test) and substantially expands the set of genes known to be varying during the day-night cycle. Additional GO terms were enriched among high-confidence day-night genes, including apoptosis and cell cycle regulation (genes peaking during the day) or fatty acid metabolism, cellular respiration, and various signaling pathways (genes peaking during the night) (Fig 3D and S9 Table). An example of high-confidence day-night gene with previously unknown circadian variation is *THRA* (S7A Fig), a thyroid hormone receptor that peaks at night in 15 tissues (including the thyroid), the expression of which was found disrupted in the hypothalamic structures of rats in constant darkness or lighting [29]. Other examples of high-confidence day-night genes present in a large number of tissues are the ribosomal protein *RPS26* (day in 16 tissues), the nuclear pore complex protein *NPIPB5* (day in 16 tissues), and the transcription corepressor *TRIM22* (day in 13 tissues) (S7B–S7D Fig).

One of the physiological signatures of the day-night rhythmicity is the sleep–wake cycle. Therefore, we next focused on a set of 254 protein-coding genes expressed in GTEx and that were previously reported to increase the risk of insomnia or to be associated with other sleep traits in humans [30] (S10 Table). From this gene set, 186 (73.2%) exhibited a significant day-night variation in at least one tissue, closely matching the genome-wide behavior across tissues (S8 Fig). Moreover, none of them belonged to the core clock, and they did not significantly overlapped with our set of high-confidence day-night genes (7 genes in common; $P = 0.49$, one-sided Fisher test) or with those annotated as circadian in the Circadian Gene DataBase in

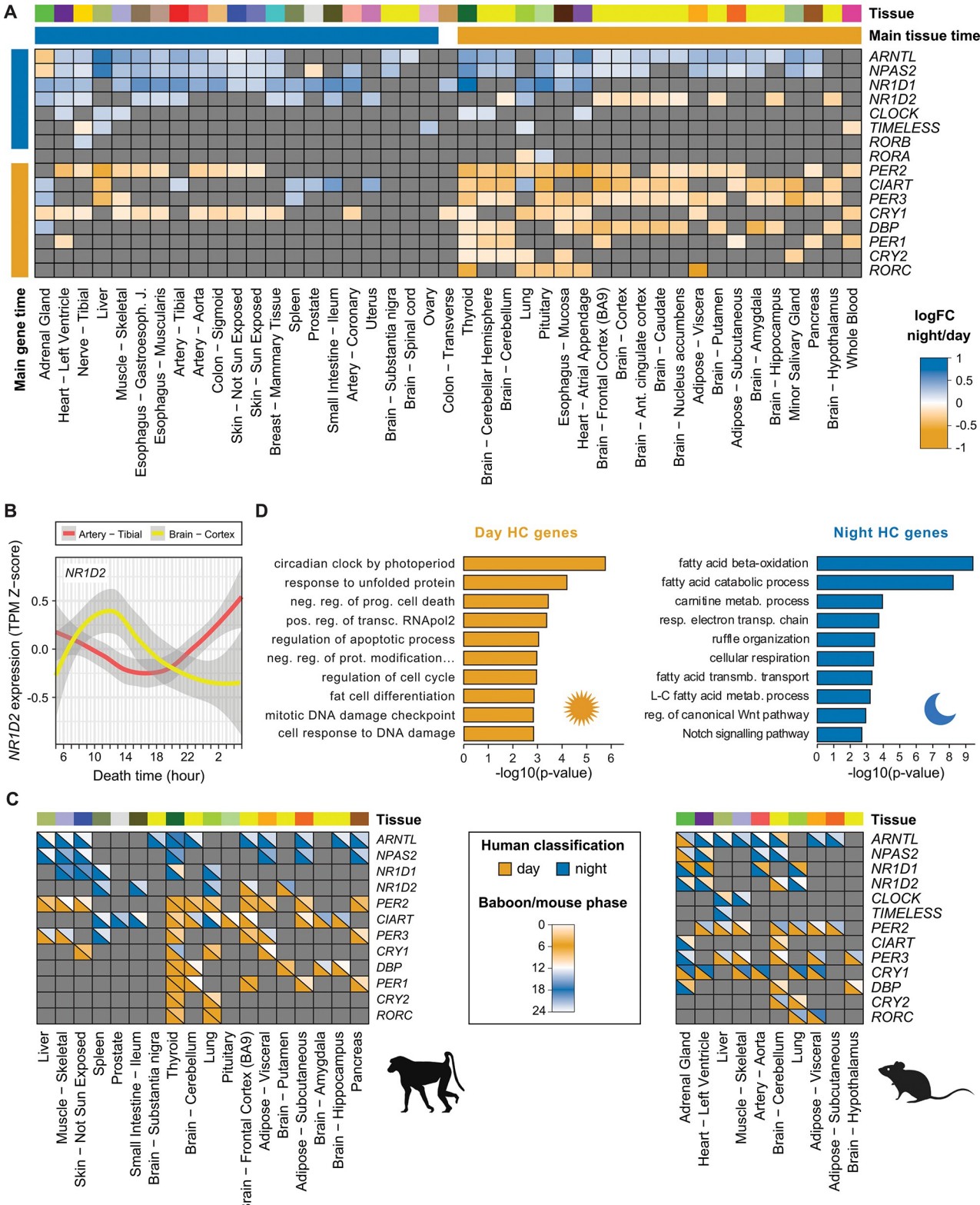

**Fig 3. Day/night differential expression of the core clock genes in human, baboon, and mouse.** (A) Day/night differential expression of the 16 core clock genes in GTEx tissues. Cells in the matrix are colored according to the log₂ fold-change obtained with the *voom-limma* pipeline, from yellow (day) to blue (night). Genes without significant effects were colored in grey. In addition, we labelled each gene ("Main gene time") as day (night) if it was up-regulated in more tissues during the day (night) than during the night (day). We labelled the tissues similarly ("Main tissue time") depending

on the number of genes that were up-regulated during the day (night) on that tissue. One gene, *RORA*, and one tissue, the aorta, were up-regulated in the same number of times during the day and the night and have not been labelled. Genes and tissues have been sorted according to (i) their main time and (ii) their number of significant effects. (**B**) Expression of *NR1D2* in the artery—tibial (red) and in the brain—cortex (yellow) GTEx samples at the time of death of the GTEx donors (in hours). Curves were obtained from the Z-score of the expression TPMs using the *geom* smooth function from *ggplot2* in R with the "loess" method. (**C**) Comparison of human and baboon (left) and human and mouse (right) orthologous core clock genes in common tissues. Only gene-tissue pairs that are significant for both compared species are shown. Cells are separated in two with (i) on the bottom left, the day-night classification of the gene in the tissue in human, and (ii) on the top right, the baboon/mouse phase obtained from Mure and colleagues [2] or Li and colleagues [26], respectively. (**D**) Top 10 enriched Gene Ontology Biological Process annotations for day and night high-confidence genes separately. See Methods for details. The data underlying this figure can be found in S1 Data.

human [28] (*P* = 0.31, one-sided Fisher test over expressed genes). These results thus suggest that genes involved in sleep traits do not appear to be particularly impacted by circadian or day-night gene expression patterns. However, four sleep genes were high-confidence day-night (S9 Fig) and annotated as circadian in the Circadian Gene DataBase [28] either in human (*PC*) or in other species (*PITPNC1* and *PDE4B* in mouse and *QSOX2* in *Arabidopsis thaliana*), pointing at a potential relevance, in specific cases, on day-night regulation of sleep behaviors. In contrast, we find a substantial overlap with genes involved in circadian rhythm sleep disorders using MalaCards [31]. Specifically, we found that 11/34 genes associated with advanced sleep phase disorder and 10/25 associated with delayed sleep phase disorder are part of the high-confidence day-night dataset. While 10 of these correspond to core clock genes, we also find one additional candidate, *CATSPER2*, in our high-confidence day-night gene set.

## Tissue-dependent seasonal variation in gene expression

Next, we analyzed the seasonal variation of gene expression. To make the analyses consistent with those of day-night patterns and to minimize the impact of GTEx reporting only the season and not the actual day of death (see Methods), we set out to identify season-specific genes, i.e., genes that are differentially more highly or more lowly expressed in one particular season versus the others. This minimizes some of the impact of measurements taken around the seasonal boundaries. Including as covariates day-night variation as well as sex, BMI, age, and postmortem interval, and using a comparable definition of differential gene expression (raw $P \leq 0.05$ and absolute $\log_2$ fold-change $\geq 0.1$; see Methods), we found that 16,408 (91.1%) of all expressed protein-coding genes were differentially expressed in at least one season in at least one tissue (hereafter seasonal genes; Supp Dataset 2 - https://doi.org/10.6084/m9.figshare.21906255.v1). There were no large differences in the number of seasonal genes across seasons, ranging from 12,026 genes in summer to 13,192 in fall. Per tissue and per season, the average number of seasonal genes were similar among seasons and comparable to day-night patterns (5.3% in spring, 4.9% in summer, 6.5% in fall, and 5.3% in winter, compared with 5.5% day-night genes), but there were more unique seasonal genes than day-night genes per tissue when all seasons were considered together (17.7%), as expected from having four versus one comparisons. The effect sizes of seasonal genes were also similar to those observed in day-night genes (S10 Fig).

As in the case of the day-night variation, the goal of this first high-sensitivity analysis was to reveal differences and similarities between patterns of seasonal variation across tissues. In this regard, in stark contrast to day-night patterns, the tissues with the highest proportion of genes showing seasonal changes included testis, skin, and multiple brain subregions, among others (Figs 4A and S11). As in the case of day-night variation, the tissue specificity of these seasonal patterns was strongly supported by randomizations (S12 Fig) and using FDR $\leq 0.1$ as a cutoff for the differential expression analysis (S13 Fig). Most of these tissues did not show a clear bias in the direction of the expression changes (i.e., the number of season-specific up- or down-

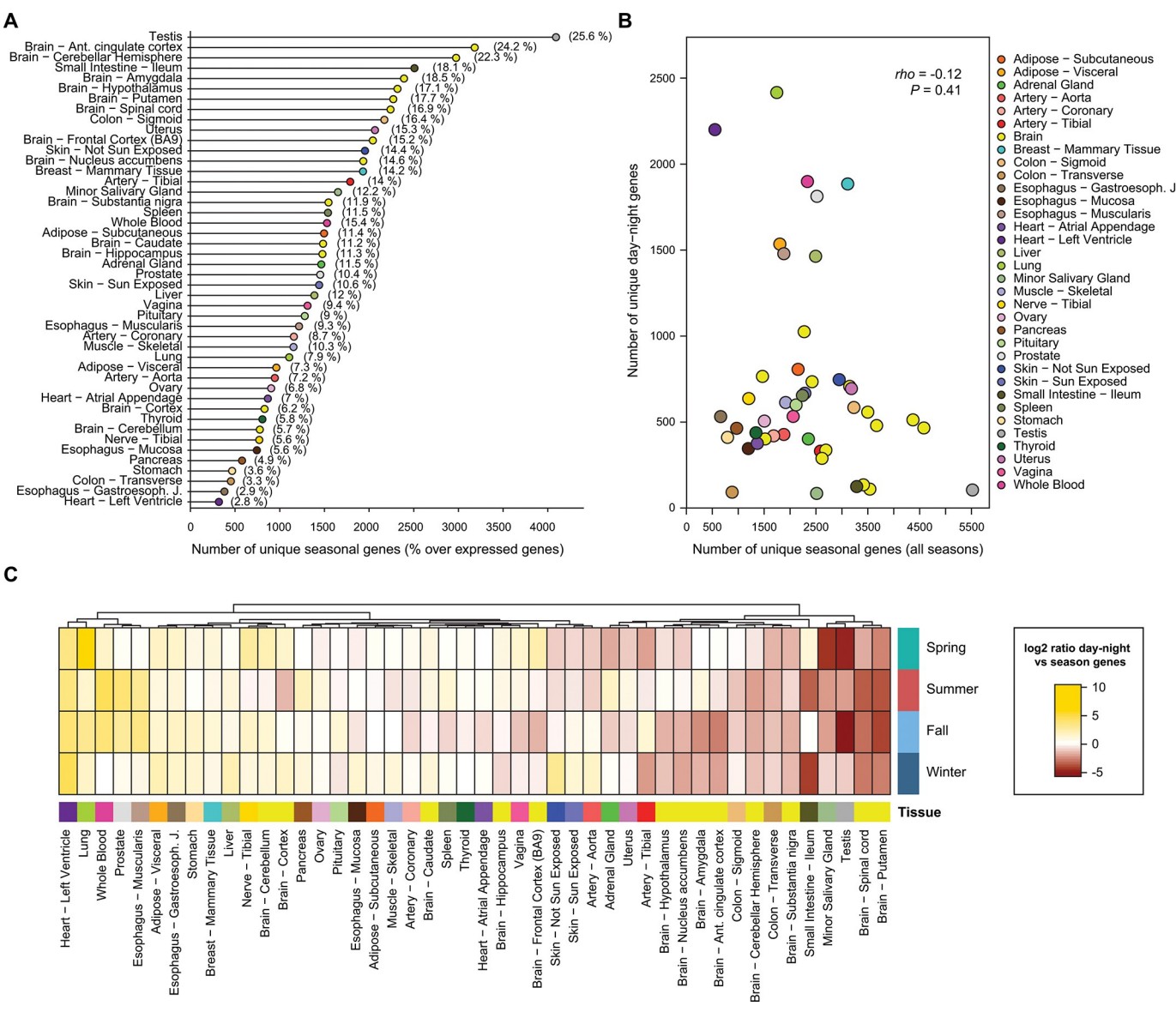

**Fig 4. Distribution of the seasonal genes in the GTEx tissues.** (**A**) Number of unique genes found as seasonal (x-axis), i.e., genes differentially expressed in at least one season when compared to the others, per tissue (y-axis). The numbers in parentheses represent the percentage of unique seasonal genes in a given tissue over the number of expressed genes in that tissue. (**B**) Number of unique seasonal vs. day-night genes per tissue. Statistics from a Spearman's correlation is shown. Tissue colors correspond to the GTEx color panel. (**C**) $\log_2$ ratio between the number of day-night genes vs. seasonal genes for each tissue and season separately. The higher the $\log_2$ value, the more day-night genes compare to the number of seasonal genes. Tissues were clustered using Euclidean distance and the Ward's method. The data underlying this figure can be found in S1 Data.

regulated genes was similar; S11 Fig and S11 Table). Testis, however, exhibited a massive gene up-regulation in fall and down-regulation in spring.

Although the overlap between seasonal and day-night genes was statistically significant for most tissues, most differentially regulated genes were unique to one or the other type of variation (S12 Table). Moreover, consistent with the distinct set of top varying tissues, the number of day-night and seasonal genes across tissues did not correlate (Spearman's rho = −0.12; Figs 4B and S14). Tissues with larger day-night than seasonal variation included various tissues from the thoracic cavity (e.g., lung and the heart's left ventricle), which may reflect changes in

heart rate and breathing patterns between day and night [32]. On the other hand, tissues with more seasonal than day-night genes included most brain subregions and gonadal tissues, likely mirroring the involvement of brain-gonadal axis in regulating seasonal physiology and behavior [33] (Fig 4C).

## Recurrent seasonal variation of gene expression across tissues

On average, seasonal genes showed seasonal expression in a number of tissues comparable to that of day-night genes: 2.5 in spring, 2.5 in summer, 3 in fall, and 2.5 winter, compared with 2.6 tissues for day-night genes. These genes also tended to be consistently up- or down-regulated across tissues. Therefore, as per day-night variation, we defined sets of high-confidence seasonal genes that varied in a consistent manner across multiple tissues for each season separately (S13 Table; see Methods). In total, we identified 1,748 unique genes: 308 in spring (138 up and 170 down), 361 in summer (158 up and 203 down), 1,072 in fall (691 up and 381 down), and 322 in winter (89 up and 233 down) (Fig 5A and S13 Table). The top enriched gene functions were largely specific for individual seasons, although some regulatory categories such as transcription and translation were shared by multiple sets (Fig 5B and S14 Table). As expected, multiple immune-related terms were enriched in fall and winter. Moreover, 122 high-confidence day-night genes were also among the high-confidence seasonal gene sets (Simpson index = 0.27; S15 Table), including three core clock genes: *PER3* (fall, summer, winter), *PER1* (winter), and *NR1D2* (fall).

An interesting example of high-confidence seasonal gene is *GLTSCR1*, a component of the SWI/SNF chromatin remodeling complex also known as *BICRA*, which increased in fall in 16 tissues (e.g., in cerebellar hemisphere; inset of Fig 5A). Similarly, *RTF1*, a component of the RNA polymerase II transcription-associated *PAF1* complex, decreased in fall in nine tissues (S15A Fig). This complex is deeply conserved across eukaryotes, and it has been described to be involved in the regulation of flowering time in plants [34,35] and to be required for induction of heat shock genes in animals [36,37]. PAF1c has been proposed to establish an antiviral state to prevent infection by incoming retroviruses: in case of infection by influenza A strain H3N2, PAF1c associates with viral NS1 protein, thereby regulating gene transcription [38]. Other examples include *C4A*, decreasing in spring in 21 tissues (S15B Fig), which localizes to the histocompatibility complex, and *KRT1*, which decreases in 24 tissues in winter (S15C Fig) and encodes a keratin gene that is a downstream effector of the corticotropin hormone release pathway [39]. *Krt1* mutations in mouse have been associated with various phenotypic defects, ranging from abnormal circulating interleukin [40] to aberrant pigmentation of the epidermis [41]. Seasonal pigment variation is well known in mammals from the northern hemisphere [42], which might suggest a conserved role of the corticotropin release pathway in pigment seasonal variation.

Finally, we focused on a set of 192 genes (1.1% of all expressed genes) that exhibited the strongest quantitative seasonal expression differences (at least a two fold-change in expression in one tissue-season pair). These were usually highly tissue specific, since only seven of these genes belonged to the set of 1,370 genes with recurrent seasonal patterns. These genes were enriched for functions related to epidermal differentiation and immunity (Fig 5C), consistent with previous results in blood cells [14]. In line with the enrichment for multiple immune-related functions, these genes also exhibited significant overlap with gene sets that have been associated with Coronavirus Disease 2019 (COVID-19), including genes whose expression changes upon Severe Acute Respiratory Syndrome Coronavirus 2 (SARS-CoV-2) infection and genes predicted to be functionally related to *ACE2* (S16 Fig), although *ACE2* itself did not show a strong seasonal pattern. In particular, 24 out of the top 200 genes among the latter

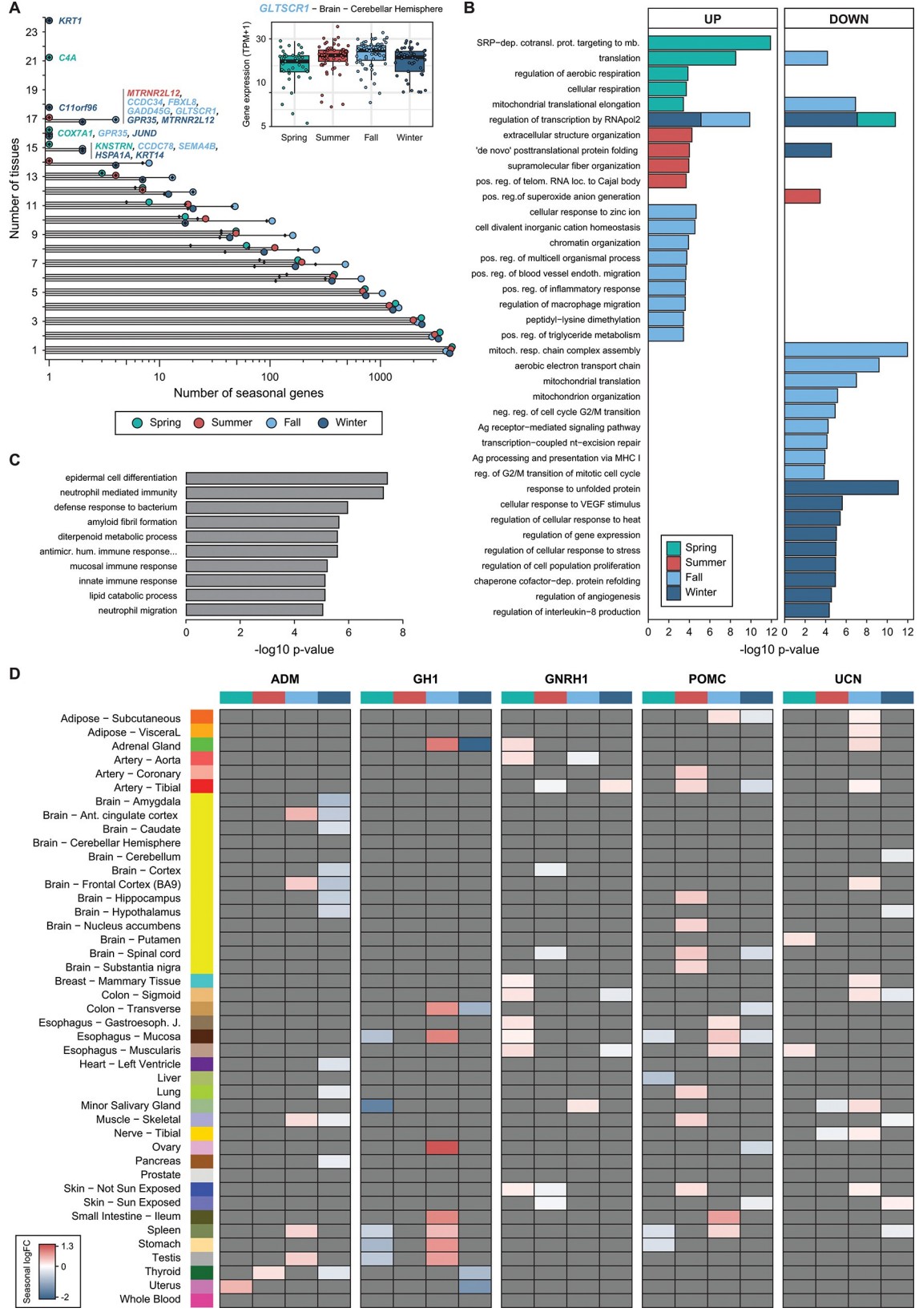

**Fig 5. Seasonal genes and associated functions.** (**A**) Distribution of the number of tissues (x-axis, $\log_{10}$ scale) in which genes were classified as seasonal (green: spring, red: summer, light blue: fall, dark blue: winter). The names of the genes found as seasonal in a large number of tissues for a given season are depicted, and the font color corresponds to their respective season. Black diamonds show the number of high-confidence seasonal genes per number of tissues. (**B**) Enriched Gene Ontology Biological Process annotations for up and down high-confidence seasonal genes separately per season (green: spring, red: summer, light blue: fall, dark blue: winter). Top terms with a raw $P < 0.005$ are shown; see Methods for details. (**C**) Gene Ontology enrichment of Biological Processes of the set of 192 strongly seasonal genes as computed by Enrichr [77,78]. (**D**) Seasonal $\log_2$ fold-change of the five hormone-coding genes included in at least one high-confidence seasonal gene set. Genes without significant effects are colored in grey. The data underlying this figure can be found in S1 Data.

predictions (12.5%) were strongly seasonal, with 20 of them being up-regulated in the intestine specifically in the winter.

## Seasonal variation of gene expression of hormone genes

Hormones [43,44] have been described to broadly regulate the body's seasonal physiology [7–9]. Thus, to explore whether genes that encode peptide hormones undergo particularly strong seasonal changes, we used a list of 73 genes with hormone-encoding capability based on Mirabeau and colleagues [45] (S16 Table). First, we compared the mRNA expression patterns of genes encoding pituitary hormones in the pituitary with the actual hormone levels, measured in blood, reported by Tendler and colleagues using medical records [43]. In general, we found a good correspondence between mRNA and hormone levels, despite the marked different approaches, especially for those peaking in summer (i.e., *POMC* [ACTH] and *TSHB* [TSH]) (S17 Fig). The main exception was *FSHB* [FSH], whose hormone levels had a low variation range and different behaviors in males and females [43].

Next, we assessed seasonality across all hormone genes and found 42 genes (58% of all hormone genes) to be seasonal in at least one tissue (S18 Fig), a significant depletion respect to the whole genome ($P = 3.27 \times 10^{-5}$, two-sided proportion test). Nevertheless, this depletion may be expected given the high degree of tissue-specific expression of most hormones (S19 Fig). Despite this, five hormone genes were included in at least one high-confidence seasonal gene set (Fig 5D). Among these, *POMC*, encoding the well-known photoperiodic hormone ACTH, was seasonal in 26 tissue-season pairs, being mainly up-regulated in summer, as in its main tissue of expression, the pituitary (see above). *POMC* expression has also been shown to be dependent on longer-term photoperiod in the Siberian hamsters [46]. Other seasonal hormones also have well-known roles, mainly in the cardiovascular system and growth: *UCN*, a corticotropin for stress response and appetite regulation [47,48], and *ADM*, which is important for vasodilation [49]. *GNRH1*, the gonadotropin-releasing hormone 1, which influences seasonal changes in other mammals [50], is seasonal in two artery tissues: the aorta and the tibial artery. Mutations in *GNRH1* have been shown to be related to ischemic heart disease, which shows a seasonal pattern [51,52]. Interestingly, leptin (*LEP*), a hormone involved in seasonal food-seeking behavior, thermoregulation [53], and obesity [54,55], was regulated only in winter in three tissues: adipose visceral, nerve, and blood (S18 Fig).

## Seasonal changes in brain cytoarchitecture

Various studies have shown seasonal histological variation in different brain regions from several mammals, including anterior cingulate cortex in shrews [56], dendritic spines in amygdala in response to short days (i.e., in fall) in white-footed mice [57], and the volume of suprachiasmatic nucleus in humans [58]. To investigate potential seasonal changes in human brain's cytoarchitecture, we used gene expression profiles of cell type–specific markers for a variety of brain cell types [59]: astrocytes, neurons, oligodendrocytes, microglia, and endothelial cells

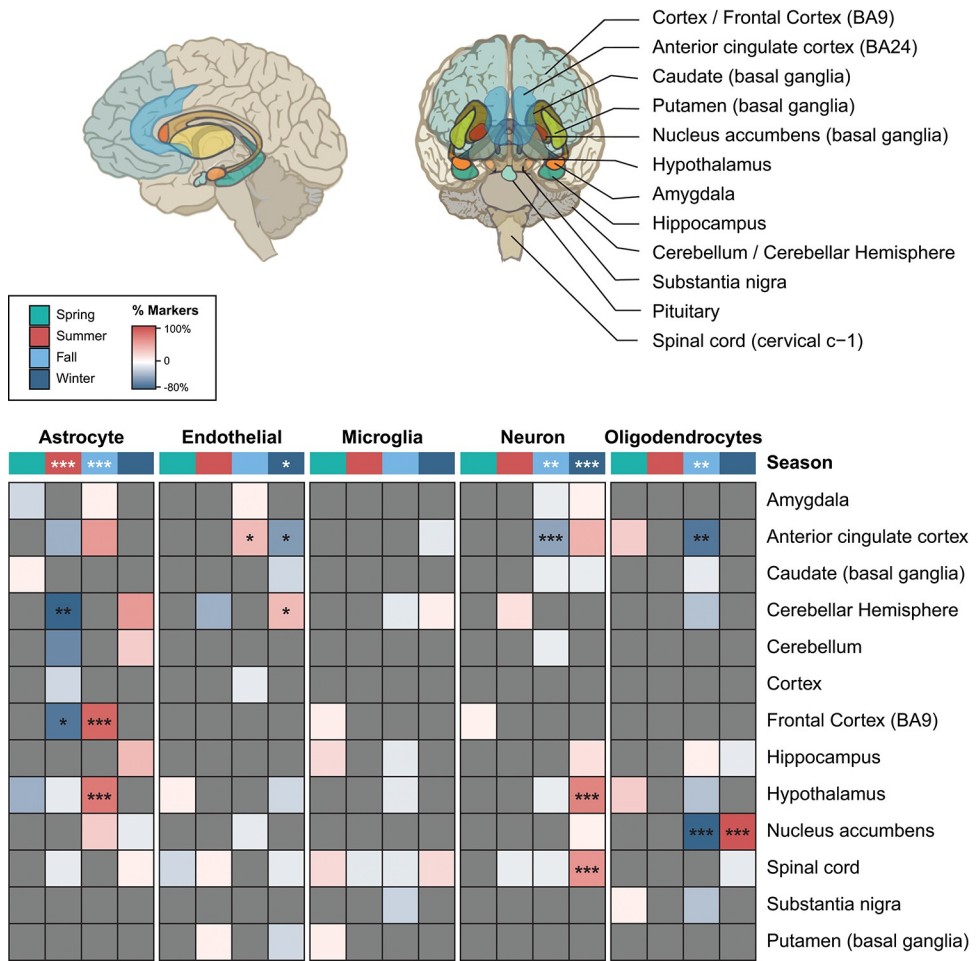

**Fig 6. Seasonal variation in cell type–specific markers across human brain regions.** Percentage of the marker genes down-regulated (blue) or up-regulated (red) for five brain cell types: astrocyte, endothelial, microglia, neuron, and oligodendrocytes, in brain subregions (depicted in the brain scheme above, downloaded and modified from the GTEx web portal). All significant markers for a cell type for a specific season in a subregion were differentially expressed in the same direction, over- or underexpressed. Markers without significant effects were colored in grey. $P$ values were obtained by randomizing the observed number of up- and down-regulated markers per cell type across regions and seasons conservatively allowing only up- or down-regulated makers for a given region–season combination (see Methods for details). *** $P < 0.001$, ** $0.001 \leq P < 0.01$, * $0.01 \leq P < 0.05$. The data underlying this figure can be found in S1 Data.

(S17 Table). We found that astrocyte markers significantly increased their expression in fall and decreased in summer (Fig 6; see Methods for statistical analysis). In particular, we observed significant increased expression of astrocyte markers in the hypothalamus and fron-tal cortex in the fall and a decrease in the cerebellum and frontal cortex in the summer. Oligo-dendrocyte markers, in contrast, tended to decrease expression in the fall, in particular, in nucleus accumbens and anterior cingulate cortex (Fig 6). In winter, all oligodendrocyte mark-ers increased their expression in nucleus accumbens. Moreover, we observed a significant increase in the expression of neuronal markers in winter, particularly in the hypothalamus and spinal cord, and a global decrease in the fall (Fig 6). Similar enrichments were obtained using a total of 627 gene markers for 16 cell type clusters from the suprachiasmatic nucleus [60] (S20 Fig). Overall, these results are thus consistent with previous histological studies in humans and other

mammals [56–58] that showed that the relative volume or cytoarchitecture of astrocytes, oligo-dendrocytes, and neurons change with the season in a subregion-specific manner.

## Potential impact of day-night and seasonal variation on drug targets

There is a growing body of evidence that circadian rhythms impact drug efficiency, including variation in the expression of targets and of transporters [61]. For example, appropriate daily timing of anticancer drugs has been suggested because of circadian regulation of cell cycle within tumors [62]. Similarly, administration of drugs to lower cholesterol levels has been proposed to adapt to the diurnal variation of this lipid [63]. Here, we have compared our set of 445 high-confidence day-night genes with the gene targets for known drugs in the Drug–Gene Interaction database (DGIdb 4.0; [64]). We found that 91 (20%) of these genes are targeted by 1,077 drugs (see S18–S20 Tables). Less is known about the impact of seasonal variation in drug administration. We have found that 307 high-confidence seasonal genes (18%) were present in DGI targeted by 2,632 drugs (see S21–S23 Tables). Interestingly, among the 18 drugs targeting at least 10 high-confidence seasonal genes, we observed 11 anticancer drugs, including borte-zomib (16 target genes), cisplatin (15), sorafenib (12), doxorubicin (12), doxorubicin (12), and ME-344 (12) (in clinical trial).

## Discussion

By leveraging the rich transcriptome data produced by the GTEx project, we have investigated the impact of the circadian and circannual cycles in the human transcriptome across multiple tissues. Ruben and colleagues [3] have also recently used the GTEx data to investigate circadian variation in gene expression; however, in that case, the time of death was inferred from the expression of known circadian genes. Here, in contrast, we use the actual time of death reported in the GTEx metadata. Even though GTEx captures a single snapshot of the tissue transcriptomes of the donors at the time of death, since this time is approximately uniformly distributed throughout the day and the year, the aggregation of the data snapshots across individuals produces trajectories that allowed us to investigate temporal variations in gene expression. Moreover, although there is an impact of the death of the individual in the transcriptome, which is tissue-specific, this can be properly controlled for [65]. The GTEx metadata, however, by making accessible only the time of the day and the season of death of the donors, makes this investigation challenging. We have addressed this limitation by discre-tizing the continuous circadian variation into day versus night and the circannual variation into seasons. This impacts the way in which we specifically formulate the questions, i.e., we do not refer to the variations reported here as circadian and circannual, but as day-night and sea-sonal, respectively, and restricts the statistical methods that can be employed to analyze the data. Thus, while ANOVA would appear the natural approach to identify seasonal changes in gene expression, the large variation in the date of death within each season decreases in prac-tice the power to detect significant changes in gene expression; therefore, we opted for focusing on a method that allows leveraging the unknown date of death and that could be more directly compared with that employed for the day-night analyses (i.e., performing pairwise differential gene expression analyses). Despite these and other caveats (effect of artificial lights, ethnic ori-gins, date of the RNA extractions, etc.), our approach was able to properly capture at least part of the real circadian and circannual transcriptional variation, since we have been able to reca-pitulate previous findings regarding day-night variation. Importantly, however, we specifically refer to it as day-night variation, since we did not strictly assess circadian oscillations. This also implies that some of this day-night variation could be due to non-circadian causes, including

diverse human activities, which could affect gene expression in a highly specific manner (e.g., food intake for digestive tissues or breathing and activity patterns for lung and heart).

We performed an initial comparison of both types of variation within and among tissues using relaxed cutoffs, finding that the effect of day-night variation in gene expression was comparable to that of the seasonal cycle, but affecting different genes and tissues. While we acknowledge that the specificity of this high-sensitivity approach might be too low to obtain reliable insights for individual genes in individual tissues, randomization analyses support that this strategy was highly robust to reveal relative differences in gene expression variation across tissues and between day-night and seasonal cycles. These comparisons showed that day-night variation was more prominent in liver, lung, heart, and upper digestive tract, reflecting the involvement of the organs of the thoracic cavity in circadian processes [66], while seasonal variation had the strongest effect in brain subregions and testis, mirroring the role of the brain-gonadal hormonal axis in regulating the physiological responses to seasonal variation [67]. Moreover, we showed that the effect of day-night and/or seasonal variation for most genes was highly tissue-specific. In the case of the day-night genes, this is in agreement with the hypothesis that one molecular clock program is present in all tissues, but the circadian processes triggered downstream are highly tissue-specific [24]. However, subsets of genes showed highly consistent up- or down-regulation during day or night or in specific seasons across multiple tissues, which we defined as high-confidence day-night or seasonal genes, respectively.

Importantly, the day-night high-confidence set was highly enriched for genes with a known circadian pattern and included most known core clock genes. In addition, various metabolic functions were enriched among day or night genes, in line with multiple reports (e.g., [68,69]). The direction of the change in gene expression for the core clock genes and other high-confidence day-night genes was highly consistent among tissues, but we found two cases in which changes in gene expression occurred unexpectedly in opposite directions between brain and non-brain tissues. This is the case of *NR1D2* and *RGS1*, in which the day-night changes in gene expression occurred in the opposite direction in the brain than in tissues from the rest of the body. This suggests that the cellular environment could modulate the interpretation of the core clock signals and produce different (or even opposite) outputs in different tissues in response to the same environmental cues.

Among all core clock genes, *ARNTL* and *NPAS2* had a differential day-night transcriptional behavior across the largest number of tissues, peaking at night time. In the circadian negative feedback loop, *ARNTL* forms a heterodimer with either *NPAS2* or its paralog *CLOCK*, positively regulating the circadian pattern [24]. Interestingly, we found that *NPAS2* shows day-night gene expression differences in a much larger number of tissues than *CLOCK* (31 versus 6 tissues), suggesting a more relevant role for *NPAS2* in the circadian modulation across tissues in human. Similarly, *NPAS2* was detected as cycling in 23 baboon tissues compared to eight tissues for *CLOCK* [2]. In contrast, mouse *Clock* and *Npas2* cycled in a similar number of tissues (eight and seven out of 12 tested tissues, respectively) [26]. Whether this difference is a lineage-specific divergence between primates and rodents, or it may be related to distinct diurnal-nocturnal habits (diurnal for human and baboon, nocturnal for mouse) will require investigation in other mammalian groups. In the latter scenario, however, the differential use of *NPAS2* and *CLOCK* could contribute to establishing the opposite circadian patterns across tissues that we observed between primates and mouse. Intriguingly, in chicken photoreceptors, *NPAS2* and *CLOCK* have been shown to positively regulate *AANAT*, the main regulator of circadian synthesis of the melatonin [70]. Despite the inverted circadian rhythm, melatonin peaks at night in both human and mouse, which suggest these regulatory interactions might not be conserved between the two lineages.

At the seasonal level, our results highlight the importance of the brain-gonadal hormonal axis. Many physiological and behavioral changes across seasons such as breeding, mating, molting, foraging, and hibernation [7,44,71,72] are known to be regulated by the endocrine system in mammals and birds, especially by the brain-thyroid and brain-gonadal axes, and can elicit different reactions depending on the receiving tissues. This is reflected in the tissue-specific seasonal variation in gene expression that we have uncovered, which affects prominently testis and many brain regions. When investigating specifically the expression patterns of genes involved in functions related to the hormonal axis, we found these to be predominantly seasonal in core tissues from the endocrine system, such as hypothalamus and pituitary. Moreover, we found that mRNA levels of various pituitary hormone genes peak in summer, in line with the reports from Tendler and colleagues [43] based on actual hormone levels in the blood obtained from medical records.

Infectious diseases are well known to have a seasonal pattern, as illustrated, for example, by the recent COVID-19 pandemics or the yearly peaks of flu infections. Consistently, we have observed a significant enrichment for immune-related functions among both the winter high-confidence gene set (Fig 5B) as well as the strongly seasonal genes (Fig 5C). Moreover, we have found a significant number of genes in various COVID-19-related pathways and associated gene sets to be strongly seasonal in a tissue-specific manner. In particular, these included genes that change expression in response to SARS-CoV-2 infection and genes predicted to be functionally related to *ACE2*, an endogenous membrane protein that mediates SARS-CoV-2 infection. These genes are likely not specific to SARS-CoV-2 infection, and their strong seasonality revealed here could shed light on the molecular mechanisms underlying the seasonality of viral infections in general.

Finally, by evaluating cell type–specific markers, our results suggest a substantial seasonal remodeling of the cytoarchitecture of certain human brain areas. We found a general increase of astrocytes in fall and winter and a decrease in summer across many subregions. These changes in the expression of cell type markers, similar to the volumetric changes described in other mammals, were subregion-specific. Interestingly, we found a reduction of neuronal markers in the anterior cingulate cortex in fall, a subregion whose neuronal soma and dendrite size shrank in the cold season in shrews [56]. Whether or not these and other putative cytoarchitectural changes are conserved in other mammals, which are their functional implications, and how they might contribute to the seasonal patterns of some psychiatric and brain diseases [10–13] need to be further investigated.

In summary, our work expands our understanding of the transcriptional impact of the physiological changes associated with the day-night cycle and, for the first time across multiple human tissues, of the seasonal cycle. Moreover, these results constitute a large resource for the community to further investigate the impact of day-night and seasonal variation in the human transcriptome. An example of the potential use of this resource is provided by our analysis on drug targets, in which we identified multiple drugs targeting high-confidence day-night or seasonal genes. Drugs targeting day-night genes could therefore be candidates to be administered in a timing-dependent dose. Similarly, it is tempting to speculate that drugs targeting genes with seasonal variation in gene expression could be administered in a seasonal dependent manner, depending on the specific tissue they need to affect. Moreover, clinical trials should also factor the season in which they are carried out.

## Methods

### RNA-seq datasets

The RNA-seq data were generated, mapped, and quantified by the GTEx consortium (GTEx v8) [15]. Tissues with less than 100 donors were discarded from the analyses (Kidney—

Medulla, Kidney—Cortex, Cervix—Ectocervix, Fallopian Tube, Cervix—Endocervix, and Bladder) as well as two cell lines (Cells—EBV-transformed lymphocytes and Cells—Cultured fibroblasts). For the Whole Blood tissue, all premortem samples were discarded from the analysis for homogeneity with the other tissues. After filtering for samples with available covariates for the differential analyses, we employed 16,151 samples from up to 46 tissues for 932 individuals. GTEx individuals are biased toward old males (median age = 55 years, 67% male). GENCODE v26 [73] was used for GTEx as well as the annotation for the protein-coding genes.

## Metacycle

The function *meta2d* from the R package MetaCycle (v.1.2) was runned for each 46 tissues separately using the default parameters. Genes with a median TPM < 1 per tissue were filtered out and the time of death of the individuals were converted into hours.

## Classification of the time of death as day or night and by season

Circadian patterns had been previously analyzed using the GTEx data, but inferring the time of death from the expression of marker genes [3]. Here, we use, instead, the actual time of death as provided by the GTEx consortium. Using this time of death, individuals had been classified between dead during the day or dead during the night if their time of death was falling into the following intervals [08:00 to 17:00] and [21:00 to 05:00), respectively (Fig 1A). Other times of death have been discarded for the day-night analysis to avoid taking into account any RNA-seq samples coming from people where the day-night status was unsure or uncertain, given the variation in day-night times along the year. We refer to these uncertain points as twilight. Using this classification, 351, 315, and 266 individuals have been classified as day, night, and twilight, respectively, of which only the 666 day and night individuals (11,527 samples) were used for differential expression for the day-night cycle. The time corresponding to day and night was manually curated from the Boston (Massachusetts) sunrise and sunset intervals during the year (using https://www.timeanddate.com/sun/usa/boston). The season of death was provided by the GTEx consortium with 190, 221, 282, and 239 donors that died in spring, summer, fall, and winter, respectively. Note that for the season, no people were discarded regarding the time of death. We also provide the information about the month in which the RNA was extracted for each season of death (S17 Table).

## Differential gene expression between day and night

Differential expression between day and night was performed separately on the 46 tissues using samples from the 666 individuals classified as day or night, ranging from 98 samples (Uterus) to 560 samples (Muscle—Skeletal). Genes were filtered per tissue, removing all genes with a median TPM < 1 over the day and night samples, leading to 18,022 protein-coding genes expressed in at least one tissue (31,530 genes including all biotypes). The analyses were run using R v3.6.1, the TMM normalization method from *edgeR* (with the *calcNormFactors* function) [74,75], and the *voom-limma* pipeline (with the *voom*, *lmFit*, and *eBayes* functions) [19,76] using default parameters. The significance of the time of death was assessed correcting for the following covariates: sex if the tissue was not sex-specific; age; BMI; the postmortem interval; the season of death. All genes with an associate $P \leq 0.05$ and an absolute $\log_2$ fold-change $\geq 0.1$ were considered as day-night. Results, including all biotypes, are available in Supp Dataset 1 - https://doi.org/10.6084/m9.figshare.21906252.v1 (including $P$ values, adjusted $P$ values, and $\log_2$ fold-changes). The number of differentially expressed genes across experiments was not affected by the number of samples (S21A Fig), and no effect was observed

regarding the ratio of day/night samples and the number of differentially expressed genes between day and night (S21B Fig).

### Definition of circadian-like genes using a sliding window approach

To identify circadian-like genes, we set up pairs of 6-hour sliding windows separated by 6 hours, i.e., [i,i+6] and [i+12,i+18] for i = 0...11, and we performed a differential gene expression analysis between both intervals for each i. This differential expression analysis was done using the same method as the one used to define our day-night genes, except that we adjusted the nominal $P$ value per gene per tissue for the number of windows ($n$ = 12) using the FDR procedure.

### Season-specific gene expression

Differential expression between seasons was performed separately on the 46 tissues using samples from the 932 individuals, ranging from 139 samples (Brain—Substantia nigra) to 789 samples (Muscle—Skeletal). Genes were filtered as described in the section "Differential expression between day and night." This resulted in a set of 18,018 protein-coding genes expressed in at least one tissue (31,517 genes including all biotypes). The effect of each season was assessed by comparing one season against all the others, leading to four differential expression analyses, one for each season (see "Differential expression between day and night" for details). This approach was taken to make the analyses more consistent with that of day-night variation (i.e., using the exact same method) and to minimize the impact of the fact that only the season, but not the actual day, of death is known in GTEx (i.e., two individuals could have died one day apart but in two different seasons, or 90 days apart but still within the same season). The covariates used were the following: time of death classified as day, night, or twilight; sex if the tissue was not sex-specific; age; BMI; postmortem interval. All genes with an associate $P \leq 0.05$ and an absolute $\log_2$ fold-change $\geq 0.1$ were considered as seasonal. Results, including all biotypes, are available in Supp Dataset 2 - https://doi.org/10.6084/m9.figshare.21906255.v1 (including $P$ values, adjusted $P$ values, and $\log_2$ fold-changes). The number of seasonal genes was not related to the number of samples per the tissue (S22A Fig), and no relation was observed between the proportion of samples available for a given season over the total and the ratio of up- and down-regulated genes. (S22B Fig).

### Testing tissues for day or night

To test if day-night genes in two tissues or more are more prone to be either day or night, we computed for each gene the ratio of day tissues over the total number of tissues for this gene. Genes will be considered to have a "consistent" ratio if their ratio is <0.25, mainly night tissues, or >0.75, mainly day tissues. For each number of tissues, from two to ten tissues, we calculate the percentage of genes that are considered to have a consistent ratio and the expected probability to be in the consistent ratio distribution following a binomial distribution with a probability of success (day tissue) equal to 0.5 and a number of draws equal to the number of genes that are day-night for this number of tissues. We then performed a binomial test for each number of tissues. The results are available at S5 Table.

### Day-night and season labels permutation

For each tissue, the day, night, and twilight labels were shuffled before any filtering, preserving the original number of each label. The twilight samples were then removed and the genes were filtered according to their median TPM (median TPM $\geq 1$). The differential gene expression

was performed as described above in "Differential gene expression between day and night." The permutation was done 1,000 times per tissue. A similar method was performed for the seasonal genes by permuting, per season, the labels of the season of interest and the other seasons before the median TPM filtering. The differential gene expression was performed as described above in "Season-specific gene expression." The permutation was done 1,000 times per tissue per season. To obtain the empirical cumulative distribution function for protein-coding genes with a non-adjusted $P \leq 0.05$ (or a non-adjusted $P \leq 0.05$ and an absolute $\log_2$ fold-change $\geq 0.1$), the proportion of time these genes have a non-adjusted $P \leq 0.05$ (or a non-adjusted $P \leq 0.05$ and an absolute $\log_2$ fold-change $\geq 0.1$) over the 1,000 permutations was computed, with all tissues pulled together. To compare the number of protein-coding genes that are defined as day-night or seasonal between the original data and the 1,000 permutations, the median per tissue (or per tissue and season) was computed (S2 and S12 Figs).

## Definition of the high-confidence gene sets

To define the five high-confidence gene sets (one for day-night genes and four for the seasonal genes), we perform three binomial tests per gene: (i) using all tissues; (ii) using only non-brain tissues; and (iii) using only brain regions. The number of draws was the number of tissues in which the gene was day-night or seasonal and the number of successes the number of tissues in which the gene was day or up. Only two of the tested genes have divergent behavior between non-brain and brain tissues: *NR1D2* (11 night non-brain tissues and 9 day brain tissues) and *RGS1* (6 day non-brain tissues and 6 night brain tissues). No divergent behavior was observed for the seasonal genes.

## Comparison of human, baboon, and mouse core clock genes

The baboon results were downloaded from Mure and colleagues [2]. We extracted the significant genes using the same threshold as Mure and colleagues ($P \leq 0.05$). Common tissues between GTEx and baboon have been manually curated (S2 Table). The mouse circadian genes were downloaded from the CirGRDB database [26], which include genes already defined as circadian, and we only selected the genes found by RNA-seq and from the publication with the PubMed ID 25349387, corresponding to the publication by Zhang and colleagues [1]. Common tissues between GTEx and the mouse tissues list had been manually curated (S7 Table). This dataset was used to compute the number of occurrences of *Clock* and *Npas2* in mouse tissues.

To perform the comparison of the core clock genes in human versus baboon, we extracted the core clock genes in both species in the 20 common tissues between the GTEx and the baboons' tissue. From this, we analyzed only genes that were significant in both species, i.e., we discarded the genes found significant in either baboon or human but not in both. The same analysis was done to compare human and mouse.

## Functional enrichment analyses

The high-confidence sets for day-night and seasonal genes, separated by day/up- and night/down-regulated genes, were used as input for the enrichment analysis tool Enrichr [77,78]. Results for GO Biological Process were retrieved and are provided fully in S9 (day-night) and S14 (seasons) Tables. To simplify visualizations in Figs 3D, 5B and 5C, we disregarded GO terms with fewer than 10 total genes. In addition, we removed redundant categories by excluding those terms sharing at least 80% of genes with a category with a lower $P$ value. For COVID-19-related gene sets, gene sets with an adjusted $P$ value greater than 0.05 were discarded.

## Circadian and hormone gene sets

The protein sequences, along with the IDs, of proteins annotated as circadian, either experimentally or by orthology, in human were downloaded from the Circadian Gene DataBase website (http://cgdb.biocuckoo.org/) [28] the 01/26/2021. The correspondence between the Ensembl protein ID and the UniProtKB ID with the Ensembl gene ID was downloaded using BioMart from Ensembl [79]. The list of hormone genes was downloaded from Mirabeau and colleagues [45]. The Ensembl peptide IDs were linked to their respective Ensembl gene ID using R and the *biomaRt* package [80,81]. The gene IDs for hormones with deprecated peptide IDs were retrieved manually using the Ensembl website (http://www.ensembl.org) and the ones that were obsolete were removed. Two hormone genes were added manually: *GH1* and *LEP*. The list is available at S16 Table.

## Statistical assessment of cell type–specific marker changes across seasons

For each cell type, we evaluated a fixed number of markers (Astrocyte: 10, Endothelial: 10, Microglia: 9, Neuron: 11, Oligodendrocytes: 10) from [59] (S17 Table). We assessed the significance of the number of up- or down-regulated markers for each cell type in each specific region-season pair (each cell in Fig 6) as well as for all regions together (top row in Fig 6). For this purpose, we first counted the total number of instances in which markers for a given cell type are up- and down-regulated across the entire set of region-season pairs (Astrocyte: 42 up, 36 down; Endothelial: 12 up, 20 down, Microglia, 9 up, 8 down; Neuron, 27 up, 12 down; Oligodendrocytes, 18 up, 27 down). Then, we performed 1,000,000 randomizations of the up and down instances across seasons and regions. We conservatively allowed only up or down markers in a given region-season pair for each cell type, since this is what we have observed in the real data (i.e., no cell type markers showed contradictory patterns in any region-season pair). To calculate the $P$ value for each region-season pair, we simply obtained the number of randomizations in which the number of up- or down-regulated markers was equal or higher than the tested region-season pair and divided it by 1 million. These $P$ values were Bonferroni corrected for multiple testing (260 tests). To calculate the $P$ value across regions, we sum all up and down instances across regions and again obtained the number of randomizations in which the number of up- or down-regulated markers was equal or higher than the sum across regions and divided it by 1 million. These $P$ values were also Bonferroni corrected for multiple testing (20 tests).

## Supporting information

**S1 Fig. Number of genes found as day-night with FDR $\leq$ 0.1.** Number of genes found as day-night, i.e., genes differentially expressed between day and night for each tissue using FDR $\leq$ 0.1. Tissues are sorted by the total number of day-night genes. The data underlying this figure can be found in S1 Data.
(PDF)

**S2 Fig. Ranks and numbers of day-night genes per tissue for the real day-night analysis and the median of 1,000 random permutations.** For each tissue, the number of day-night genes from the real data (GTEx, circle) or the median number of day-night genes over the 1,000 random permutations (permutation, diamond) was computed (y-axis). The tissues were ordered according their number of day-night genes (x-axis), independently for the GTEx and the permutation dataset. DEGs, differentially expressed genes. The data underlying this figure can be found in S1 Data.
(PDF)

**S3 Fig. Correspondence between circadian-like and day-night genes.** (**A**) Number of differentially expressed genes between sliding pairs of 6-hour time windows, defined as [i,i+6) and [i+12,i+18) for i = 0. . .11. Each step i corresponds to 1 hour. (**B**) Scatter plot showing the number of genes that are differentially expressed between at least one pair of time windows ("circadian-like" genes) and of day-night genes per tissue. The data underlying this figure can be found in S1 Data.
(PDF)

**S4 Fig. Absolute *log$_2$* fold-change (or effect size) of day-night genes.** Each core clock gene in the high-confidence day-night set is plotted separately (green boxplots). The rest of the genes are pooled together in "Others—Core clock" if they are in the list of the core clock genes, in "Others—High-confidence set" if they are part of the high-confidence set, or in "Others" otherwise. The data underlying this figure can be found in Supp Dataset 1 (https://doi.org/10.6084/m9.figshare.21906252.v1).
(PDF)

**S5 Fig. Contrasting *NR1D2* circadian behaviors in brain and other tissues.** TPM values for *NR1D2* in the artery—tibial (left) and in the brain—cortex (right). The colors of the dots represent the classification of the individuals according to the time of death of the donor: during the day (yellow), during the night (blue), or in-between for twilight (grey). The samples classified as twilight have been discarded for the day-night analysis. The "circadian" curve was created using the *geom_smooth* function from *ggplot2* in R with the "loess" method. The data underlying this figure can be found in S1 Data.
(PDF)

**S6 Fig. Classification of genes in the high-confidence day-night set.** For each number of tissues in which a gene was identified as day-night, the percentage of genes that are consistently up-regulated either during day or during night for (i) all tissues taken together ($P \leq 0.05$ for the binomial test on all tissues; All tissues, green); (ii) in non-brain and in brain tissues separately ($P \leq 0.05$ for the binomial test on non-brain and brain tissues and $P > 0.05$ for all tissues; In separated tissue types, blue); (iii) in non-brain tissues only ($P \leq 0.05$ only for the non-brain tissues test; In non-brain only, orange); (iv) in brain tissues only ($P \leq 0.05$ only for the brain tissues test; In brain only, yellow); and (v) inconsistent between day and night tissues ($P > 0.05$ for all binomial tests; Not in high-confidence set, black). See Methods for details. The data underlying this figure can be found in S8 Table.
(PDF)

**S7 Fig. Examples of day-night genes.** Expression values (TPM) values for three high-confidence day-night genes at the time of death of the GTEx donors: (**A**) *THRA* in thyroid, (**B**) *RPS26* in the whole blood, (**C**) *NPIPB5* in the esophagus-gastroesophageal junction, and (**D**) *TRIM22* in the adipose-visceral (omentum). The colors of the dots represent the classification of the individuals according to their time of death of the donor: during the day (yellow), during the night (blue), or in-between for twilight (grey). The samples classified as twilight have been discarded for the day-night analysis. The "circadian" curve was created using the *geom_smooth* function from *ggplot2* in R with the "loess" method. The data underlying this figure can be found in S1 Data.
(PDF)

**S8 Fig. Day-night variation of sleep associated genes.** Number of sleep day-night genes (y-axis) vs. total day-night genes (x-axis) per tissue. The data underlying this figure can be found

in S1 Data.
(PDF)

**S9 Fig. Example of day-night sleep genes.** Expression values (TPMs) for the four sleep-related genes in the day-night high-confidence gene set and annotated in the Circadian Gene Data-Base: (**A**) *PC* in cerebellar hemisphere, (**B**) *PITPNC1* in anterior cingulate cortex, (**C**) *PDE4B* in subcutaneous adipose, and (**D**) *QSOX2* in pituitary at the time of death of the GTEx donors. The colors of the dots represent the classification of the individuals according to the time of death of the donor: during the day (yellow), during the night (blue), or in-between for twilight (grey). The samples classified as twilight have been discarded for the day-night analysis. The "circadian" curve was created using the *geom_smooth* function from *ggplot2* in R with the "loess" method. The data underlying this figure can be found in S1 Data.
(PDF)

**S10 Fig. Effect size of day-night and seasonal variation in gene expression.** Boxplots showing the absolute $\log_2$ fold-change (or effect size) of day-night genes (yellow) or seasonal genes (red: summer, green: spring, light blue: fall, dark blue: winter). The data underlying this figure can be found in Supp Datasets 1 and 2 (https://doi.org/10.6084/m9.figshare.21906252.v1 and https://doi.org/10.6084/m9.figshare.21906255.v1, respectively).
(PDF)

**S11 Fig. Landscape of seasonal gene expression variation by tissue.** Number of overexpressed (Season up, right side) and underexpressed (Season down, left side) seasonal genes per tissue in GTEx (bottom axis) for each season, spring (**A**), summer (**B**), fall (**C**), and winter (**D**). The tissues were ordered according to the total number of seasonal genes per season. The red dot represents the $\log_2$ ratio between the number of over and under genes (top axis). The data underlying this figure can be found in S1 Data.
(PDF)

**S12 Fig. Ranks and numbers of seasonal genes per tissue for the real seasonal analyses and the median of 1,000 random permutations.** For each tissue, the number of spring (**A**), summer (**B**), fall (**C**), or winter (**D**) genes from the real data (GTEx, circle) or the median number of genes differentially expressed in each season over the 1,000 random permutations (permutation, diamond) was computed (y-axis). The tissues were ordered according their number of seasonal genes (x-axis), independently for the GTEx and the permutation dataset. DEGs, differentially expressed genes. The data underlying this figure can be found in S1 Data.
(PDF)

**S13 Fig. Landscape of seasonal gene expression variation by tissue using FDR $\leq$ 0.1.** Number of unique genes found as seasonal (x-axis), i.e., genes differentially expressed in at least one season when compared to the others using FDR $\leq$ 0.1 as cutoff, per tissue (y-axis). The numbers in parentheses represent the percentage of unique seasonal genes in a given tissue over the number of expressed genes in that tissue. The data underlying this figure can be found in S1 Data.
(PDF)

**S14 Fig. Number of unique seasonal vs. day-night genes per tissue using FDR $\leq$ 0.1.** Statistics from a Spearman's correlation is shown. Tissue colors correspond to the GTEx color panel. The data underlying this figure can be found in S1 Data.
(PDF)

**S15 Fig. Examples of seasonal genes.** Boxplot of the TPM values on a $\log_{10}$ scale of three top seasonal genes in a tissue where they are differentially expressed: *RTF1*, underexpressed in fall

(**A**), *C4A* underexpressed in spring (**B**), and *KRT1* underexpressed in winter (**C**). The data underlying this figure can be found in S1 Data.
(PDF)

**S16 Fig. Enrichment in COVID-19-related genes of the 192 strongly seasonal genes as computed by Enrichr.** *ACE2* Geneshot AutoRIF ARCHS4 predictions were obtained by Geneshot, combining genes previously published to be associated with *ACE2*, as well as genes predicted to be associated with ACE2 based on data integration from multiple sources, including coexpression matrices based on RNA-seq data and others.
(PDF)

**S17 Fig. Comparison of mRNA levels of hormone genes and plasma hormone levels from Tendler and colleagues.** Left: Median mRNA expression levels for pituitary hormone genes in each season from GTEx. Centre: Average of male and female median values for hormones from Tendler and colleagues [43] obtained from Clalit medical records. Right: Average values group by season (Winter: Jan, Feb, Mar; Spring: Apr, May, Jun; Summer: Jul, Aug, Sep; Fall: Oct, Nov, Dec). Genes/hormones correspond to *POMC*/ACTH (**A**), *TSHB*/TSH (**B**), *LHB*/LH (**C**), *FSHB*/FSH (**D**), and *GH1*/GH (**E**). The data underlying this figure can be found in S1 Data.
(PDF)

**S18 Fig. Landscape of the seasonal variation of human hormone genes.** Seasonal $log_2$ foldchange for the 42 hormone genes in the 45 tissues (all except vagina) with at least one hormonal gene showing seasonal expression. The data underlying this figure can be found in S1 Data.
(PDF)

**S19 Fig. Summary of the seasonal variation of human hormone genes.** (**A**) For each tissue, the number of seasonal hormone genes with a median expression TPM $\geq$ 5. (**B**) For each seasonal hormone gene, the number of tissues for which the gene had a median expression TPM $\geq$ 5. The data underlying this figure can be found in S1 Data.
(PDF)

**S20 Fig. Seasonal variation in cell type–specific markers from Wen and colleagues across human brain regions.** Percentage of the marker genes down-regulated (blue) or up-regulated (red) for major cell types identified by Wen and colleagues [60] in brain subregions (depicted in the brain scheme in Fig 6). Markers without significant effects were colored in grey. The data underlying this figure can be found in S1 Data.
(PDF)

**S21 Fig. Correlations between day-night variation and sample size.** (**A**) Number of day-night genes per number of samples. (**B**) Ratio of the number of day over the number of night genes vs. the ratio of day over night samples. The data underlying this figure can be found in S1 Data.
(PDF)

**S22 Fig. Correlations between seasonal variation and sample size.** (**A**) Number of seasonal genes vs. number of seasonal samples per season. (**B**) Ratio of the number of genes found overexpressed over genes found underexpressed per season vs. the ratio of the samples from a specific season over all samples. The data underlying this figure can be found in S1 Data.
(PDF)

**S1 Table. Day-night and MetaCycle results.** For each tissue, number of genes found by the day-night analysis, MetaCycle, and the genes they found in common.
(XLSX)

**S2 Table. Homologous tissues for human and baboon.** Compared homologous tissues between the human GTEx dataset and baboon dataset from Mure and colleagues [2].
(XLSX)

**S3 Table. Summary statistics of day-night expression analysis.** For each tissue, number of genes with differentially up-regulated expression during the day or during the night, percentage of expressed genes with a day or night expression, number of expressed genes, and log2 of the ratio day vs. night up-regulation. Corresponding to S2A Fig.
(XLSX)

**S4 Table. Phase classification of the genes missed by the day-night analysis but found by MetaCycle.** For each tissue, for the genes that are found by MetaCycle but missed by the day-night analysis, the number of genes that are classified as day, night, twilight, or within 1 hour of the twilight based on the predicted phase of MetaCycle.
(XLSX)

**S5 Table. Binomial test of the number of day-night genes in consistent bins.** For genes that were identified as day-night in a given number of tissues (from two to ten), it is shown the number of genes that are in consistent bins, i.e., genes with a ratio of day tissues over the total number of tissues $< 0.25$ or $>0.75$, the total number of genes, and the $P$ value from the binomial test obtained using the expected ratio from a random distribution for that number of tissues as the hypothesized probability of success ("Probability of being in consistent bins"). See Methods for details.
(XLSX)

**S6 Table. List of core clock genes.**
(XLSX)

**S7 Table. Homologous tissues for human and mouse.** Compared homologous tissues between the human GTEx dataset and mouse dataset from Li and colleagues [26].
(XLSX)

**S8 Table. Day-night genes and their associated binomial $P$ value.** For each gene, the number of tissues for which the gene is day or night, for (i) all tissues, (ii) non-brain tissues, and (iii) brain tissues. For each of the previous tissue classification, a binomial $P$ value is associated to test if the gene has a statistically significant preference to be highly expressed during the day or the night. Genes with a $P \leq 0.05$ for at least one of the tissue classifications are part of the high-confidence day-night gene sets. Note that for a gene to reach a $P \leq 0.05$ it needs to have been identified day-night in at least 6 tissues. See Methods for details.
(XLSX)

**S9 Table. Gene Ontology enrichment results for the high-confidence day-night genes for day and night genes separately.** Enrichment performed with EnrichR. See Methods for details.
(XLSX)

**S10 Table. Genes associated with sleep traits.** List of 254 protein-coding genes expressed in GTEx and that were previously reported to increase the risk of insomnia or to be associated with other sleep traits in humans.
(XLSX)

**S11 Table. Summary statistics of the seasonal expression analysis.** For each tissue and season, number of genes with differentially up- or down-regulated expression, percentage of expressed genes with up- or down-regulation in that given tissue and season, number of expressed genes, and log2 of the ratio up/down.
(XLSX)

**S12 Table. Overlap between day-night and seasonal variation.** For each tissue, number of genes with day-night expression, seasonal expression, overlap between the two sets and $P$ value for the overlap (Fisher exact test with and without Benjamini–Hochberg multiple testing correction).
(XLSX)

**S13 Table. Seasonal genes and their associated binomial $P$ value.** For each season and seasonal gene, the number of tissues for which the gene is up- or down-regulated, for (i) all tissues, (ii) non-brain tissues, and (iii) brain tissues. For each of the previous tissue classification, a binomial $P$ value is associated to test if the gene has a statistically significant preference to be up- or down-regulated in that season. Genes with a $P \leq 0.05$ for one of the tissue classifications are part of the high-confidence seasonal gene sets. Note that for a gene to reach a $P \leq 0.05$, it needs to have been identified as differentially expressed in that season in at least 6 tissues. See Methods for details.
(XLSX)

**S14 Table. Results from the Gene Ontology enrichment analysis for the high-confidence seasonal genes per season and for up- and down-regulated genes separately.** Enrichment performed with EnrichR. See Methods for details.
(XLSX)

**S15 Table. Overlap between day-night and seasonal high-confidence gene sets.** For each gene, the number of tissues in which it is up-regulated in day and/or night as well in specific seasons is provided.
(XLSX)

**S16 Table. List of human hormone genes.** List of 62 genes with hormone-encoding capability based on Mirabeau and colleagues [45].
(XLSX)

**S17 Table. Cell type–specific markers.** List of genes serving as cell type–specific markers in the human brain from McKenzie and colleagues [59].
(XLSX)

**S18 Table. Interactions between high-confidence day-night genes and drugs.** List of drug-target day-night gene pairs. The number of tissues in which the day-night gene is up-regulated during the day and/or night is provided.
(XLSX)

**S19 Table. Drugs targeting high-confidence day-night genes.** List of drugs ranked by the number of targeted high-confidence day-night genes.
(XLSX)

**S20 Table. High-confidence day-night genes targeted by drugs.** List of high-confidence day-night genes ranked by the number of targeting drugs. The number of tissues in which the day-night gene is up-regulated during the day and/or night is provided.
(XLSX)

**S21 Table. Interactions between high-confidence seasonal genes and drugs.** List of drug-target seasonal gene pairs. The number of tissues in which the seasonal gene is up- or down-regulated in each season is provided.
(XLSX)

**S22 Table. Drugs targeting high-confidence seasonal genes.** List of drugs ranked by the number of targeted high-confidence seasonal genes.
(XLSX)

**S23 Table. High-confidence seasonal genes targeted by drugs.** List of high-confidence seasonal genes ranked by the number of targeting drugs. The number of tissues in which the seasonal gene is up- or down-regulated in each season is provided.
(XLSX)

**S24 Table. Month of RNA extraction and season of death.** The month of the RNA extractions of the samples and the season of death of the individuals.
(XLSX)

**S1 Data. Raw data used to generate each figure panel.**
(XLSX)

## Acknowledgments

We thank the donors and their families for their generous gifts of biospecimens to the GTEx research project; the Genomics Platform at the Broad Institute for data generation. We also thank Manuel Muñoz-Aguirre and Thomas Derrien for their feedback on the manuscript, Diego Garrido Martín for his help on side analysis, Claudia Vivori and Vanessa Vega Méndez for providing help with graphical presentations. Animal silhouettes were downloaded from http://phylopic.org.

## Author Contributions

**Conceptualization:** Valentin Wucher, Reza Sodaei, Roderic Guigó.

**Data curation:** Valentin Wucher, Reza Sodaei.

**Formal analysis:** Valentin Wucher, Reza Sodaei.

**Funding acquisition:** Roderic Guigó.

**Investigation:** Valentin Wucher, Reza Sodaei, Raziel Amador.

**Methodology:** Valentin Wucher, Reza Sodaei.

**Project administration:** Roderic Guigó.

**Resources:** Roderic Guigó.

**Software:** Valentin Wucher, Reza Sodaei.

**Supervision:** Manuel Irimia, Roderic Guigó.

**Visualization:** Valentin Wucher, Manuel Irimia.

**Writing – original draft:** Valentin Wucher, Reza Sodaei, Manuel Irimia.

**Writing – review & editing:** Manuel Irimia, Roderic Guigó.

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
