## [Editor Report · Decision Letter 0]

23 Jun 2022

Dear Dr Irimia, 

Thank you for submitting your manuscript entitled "Day-night and seasonal variation of human gene expression across tissues" for consideration as a Research Article by PLOS Biology, and please let me apologize for our delay in sending you an initial decision.

Your manuscript has now been evaluated by the PLOS Biology editorial staff as well as by an academic editor with relevant expertise and I am writing to let you know that we would like to send your submission out for external peer review.

Once your full submission is complete, your paper will undergo a series of checks in preparation for peer review. After your manuscript has passed the checks it will be sent out for review. To provide the metadata for your submission, please Login to Editorial Manager (https://www.editorialmanager.com/pbiology) within two working days, i.e. by Jun 27 2022 11:59PM.

Kind regards,

Lucas

Lucas Smith, Ph.D.

Associate Editor

PLOS Biology

lsmith@plos.org

---

## [Decision Letter · Decision Letter 1]

28 Jul 2022

Dear Dr Irimia,

Thank you for your patience while your manuscript "Day-night and seasonal variation of human gene expression across tissues" was peer-reviewed at PLOS Biology. It has now been evaluated by the PLOS Biology editors, an Academic Editor with relevant expertise, and by several independent reviewers. 

In light of the reviews, which you will find at the end of this email, we would like to invite you to revise the work to thoroughly address the reviewers' reports.

As you will see, while the reviewers find the analysis performed here interesting, and appreciate the importance of the topic, they have also raised a number of important concerns which will need to be thoroughly addressed before we can consider your manuscript further for publication. While we expect the revised manuscript to carefully address all of the reviewers’ comments, I would like to emphasize that it would be particularly important for your revision to provide the additional analyses requested by Reviewer 3, as we feel that strengthening the conclusions of the study would be essential for further consideration.

In addition to addressing the reviewer concerns, the Academic Editor has also commented that reference [42] (Tendler et al, 2021) is currently weakly cited. We feel that part of the strength of the study is that it may provide some mechanistic aspects to the phenomenon described in [42], and would encourage you to add a bit more discussion about how your findings relate to that manuscript.

As a last note, Reviewer 2 has commented that the study may be more appropriate for PLOS One and has suggested that we reject the study. However, after discussion with the Academic Editor, we do not share this specific concern, as we think that, if properly strengthened, that the manuscript would be a strong resource for the community and would be within the scope of our journal.

We would therefore like to invite you to revise the work to thoroughly address the reviewers' reports. Given the extent of revision needed, we cannot make a decision about publication until we have seen the revised manuscript and your response to the reviewers' comments. Your revised manuscript is likely to be sent for further evaluation by all or a subset of the reviewers.

**IMPORTANT - SUBMITTING YOUR REVISION**

*Re-submission Checklist*

*Published Peer Review*

*PLOS Data Policy*

*Blot and Gel Data Policy*

Sincerely,

Lucas

Lucas Smith, Ph.D.

Associate Editor

PLOS Biology

lsmith@plos.org

REVIEWS:

Reviewer #1: The authors exploit the valuable resource of GTEx expression profiles from human tissues with respect to day-night differences and seasonal variations. There are related studies such as Ruben 2019 and Talamanca (BioRxiv) but Wucher et al. apply specific bioinformatics methods to mine the exciting database. Instead of extractions of circadian phases and amplitudes as in the other studies, they analyze expression levels between day and night and across seasons with established statistical tools. They find novel hormonal and immunological regulations and seasonal variations of brain regions.

The analysis is based on the time of death. Is there information available regarding the time span between death and RNA-extraction? Other groups (Ruben, Talamanca) exploit the core clock genes to define an internal time frame via CYCLOPS or Per3 expression. Is it possible to study phase shifts between time of death and these phase markes? Even though this might be an interesting topic for the discussion the analysis in the manuscript uses coarse-grained definitions of day and night and seasons and thus I do not expect major effects on the precise extraction of intrinsic time.

The authors apply the most appropriate and up-to-date bioinformatics methods: MetaCycle, Lomb-Scargle, voom/limma, permutations of data, multiple testing corrections. I found it helpful that different levels of sensitivity have been used. 

Many results are consistent with previous findings of Ruben et al. Interestingly, NR1D2 has different behavior in brain and other tissues. This is relevant since NR1D2 is used often in body-time studies (see e.g. Witenbrink 2018). Another interesting finding is the association with insomnia genes.

To me, the seasonal overexpression studies are really novel. Interestingly, they seem weakly related to diurnal rhythms (see Fig. 4B). As expected there are interesting relations to the yearly rhythms of specific hormones.

It is intriguing to see differentials of Clock and Npas2 in rodents and humans. It is known that light and feeding are out-of-phase but melatonin is peaking in the dark in both, in nocturnal and diurnal animals. Is there any link between melatonin and Clock or Npas2 known?

Reviewer #2: Although I think the authors has done important research and I have no doubt of the validity of the results, which do deserve publishing, in my opinion the results are too vague for PLOS Biology and would be more appropriate for PLOS1. 

When trying to summarise in my head what are the most important conclusions of this research, the best I could do was the following 

1. There is a day/night and seasonal variation of gene expression, but different genes are expressed (variably on these factors) in different tissues.

2. The genes that are variably expressed with high confidence between day and night include known circadian genes, but there are others previously unknown; some of them, including the ones that are variably expressed in many tissues are hard to explain (such as ribosomal protein RPS26).

3. There is seasonal variation in hormonal and brain genes. 

Have I missed something important? If not, I think that the authors will agree that this is a bit thin for PLOS Biology. 

But moreover, I think there is one flaw that needs to be addressed before this can be considered ffurther. As far as I was able to tell, no attempt has been made to separate the direct circadian variation from indirect effects of human day/night activities. For instance, one of the conclusions is that many genes have day/night variability in heart, lung and digestive tract. But humans are typically more active during the day, with higher heart rates and also more exercised lung. The eating patters too are different, which may be expected to affect digestive tract. I do not know if heart gene expression is known to depend on the activity level, but I would be somewhat surprised if it was not. In any case, I think the authors need to look into these possible secondary effects. (At least in one instance they have done this for seasonal variation - there are more infections during winter, which affects the immune system). Possibly there is no sufficient metadata in GTEx to do this in-depth, but at least this has to be discussed. Perhaps, some of the genes they found can be explained by such secondary effects? On a related matter, perhaps the authors could also discuss some of the genes that are difficult to explain, such as RPS26. Can this be an artifact? 

Reviewer #3: Attached!

---

## [Editor Report · Decision Letter 2]

9 Dec 2022

Dear Dr Irimia,

Thank you for your patience while we considered your revised manuscript "Day-night and seasonal variation of human gene expression across tissues" for publication as a Research Article at PLOS Biology. This revised version of your manuscript has been evaluated by the PLOS Biology editors and by the Academic Editor, who feels the manuscript has been greatly improved and is satisfied by the changes made in response to the last round of review. 

Based on our Academic Editor's assessment of your revision we are likely to accept this manuscript for publication. However, before we can editorially accept your manuscript, we need you to address the following two editorial requests in another round of revision. 

**EDITORIAL REQUESTS: 

1) After some discussion within the team, we think your manuscript would be best suited for our "Methods and Resources" article type. When you resubmit your manuscript, please change the article type accordingly. For more information on Resource articles, see here: https://journals.plos.org/plosbiology/s/what-we-publish#loc-methods-and-resources-articles

Note that we do not require all raw data. Rather, we ask that all individual quantitative observations that underlie the data summarized in the figures and results of your paper be made available in one of the following forms.

I see that you have provided several supplemental tables, including summary statistics of day-night and or seasonal expression analyses which is great. However it is not immediately apparent to me that these excel files contain all the data needed to be in compliance with our data policy (apologies if I missed some of the relevant data). 

What we are looking for is supplementary files that contain the individual numerical values that underlie the summary data displayed in your figures. Please ensure that your supplemental excel files contain this for the following figure panels as they are essential for readers to assess your analysis and to reproduce it:

Fig 2A-D; Fig 3A-C; Fig 4B-C; Fig 5A,D; Fig 6; Fig S1; Fig S2; Fig S3A-B; Fig S4; Fig S5; Fig S6; Fig S7A-D; Fig S8; Fig S9A-D; Fig S10; Fig S11A-D; Fig S12; Fig S13; Fig S14; Fig S15A-C; Fig S17; Fig S18; Fig S19; Fig S20; Fig S21A-B; Fig S22A-B;

>>I apologize if some of this data is already contained within the supplemental tables that you have provided, and if I missed it. To aid us (and the reader) in finding this underlying data, please ensure that figure legends in your manuscript include information on where the underlying data can be found for that figure, and ensure your supplemental data file/s has a legend. For example, to each figure legend you can add the sentence "the data underlying this figure can be found in ___"

>>Please make sure to update your Data Statement in the submission system accurately describes where your data can be found. For example, you could change it to read: All GTEx open-access data are available on the GTEx Portal (https://gtexportal.org/home/datasets). All GTEx protected data are available via dbGaP (accession phs000424.v8). Access to the raw sequence data is now provided through the AnVIL platform (https://gtexportal.org/home/protectedDataAccess). Drugs and their target genes weredownloaded from the DGI database (https://dgidb.org/downloads, version 2022-Feb). All other relevant data are within the paper and its Supporting Information files

We expect to receive your revised manuscript by January 3rd. (Please note that the PLOS Biology editorial office will be closed for the Holidays, during the last two weeks of December) 

*Published Peer Review History*

*Press*

Sincerely,

Lucas

Lucas Smith, Ph.D.

Associate Editor,

lsmith@plos.org,

PLOS Biology

---

## [Editor Report · Decision Letter 3]

3 Jan 2023

Dear Dr Irimia,

Thank you for the submission of your revised Methods and Resources article "Day-night and seasonal variation of human gene expression across tissues" for publication in PLOS Biology. Apologies for our delay in sending you a decision - much of the PLOS Biology editorial team has been out of the office over the last two weeks for the Holidays. We have now had a chance to evaluate your revised study and we are satisfied by the changes you made in response to our editorial requests (and we have gone ahead and updated the article type to 'Methods and Resources'). Therefore, on behalf of my colleagues and the Academic Editor, Martha Merrow, I am pleased to say that we can in principle accept your manuscript for publication, provided you address any remaining formatting and reporting issues. These will be detailed in an email you should receive within 2-3 business days from our colleagues in the journal operations team; no action is required from you until then. Please note that we will not be able to formally accept your manuscript and schedule it for publication until you have completed any requested changes.

PRESS

Sincerely, 

Lucas Smith, Ph.D.

Associate Editor

PLOS Biology

lsmith@plos.org